# Urachal Carcinoma, An Unusual Possibility of Hematuria; Case Report and Literature Review

**DOI:** 10.3390/diagnostics12081892

**Published:** 2022-08-04

**Authors:** Răzvan Călin Tiutiucă, Alina Ioana Năstase Pușcașu, Elena Țarcă, Nicoleta Stoenescu, Elena Cojocaru, Laura Mihaela Trandafir, Viorel Țarcă, Dragoș-Viorel Scripcariu, Mihaela Moscalu

**Affiliations:** 1Surgical Department, Iacob Czihac Military Emergency Clinical Hospital, 700483 Iasi, Romania; 2Regional Institute of Oncology, 700483 Iasi, Romania; 3Department of Surgery II-Pediatric Surgery, “Grigore T. Popa” University of Medicine and Pharmacy, 700115 Iasi, Romania; 4Department of Morphofunctional Sciences I, “Grigore T. Popa” University of Medicine and Pharmacy, 700115 Iasi, Romania; 5Department of Mother and Child, “Grigore T. Popa” University of Medicine and Pharmacy, 700115 Iasi, Romania; 6Department of Communication Sciences, Apollonia University, 700613 Iasi, Romania; 7Surgical Department, “Grigore T. Popa” University of Medicine and Pharmacy, 700115 Iasi, Romania; 8Department of Preventive Medicine and Interdisciplinarity, “Grigore T. Popa” University of Medicine and Pharmacy, 700115 Iasi, Romania

**Keywords:** hematuria, urachal adenocarcinoma, urachal abnormalities

## Abstract

Urachal cancer is very rare, accounting for only 0.5–2% of bladder-associated malignancies and 0.01% of all cancers in adults. It has an insidious appearance, an aggressive behavior and a poor prognosis. The most common symptoms are hematuria and the presence of a palpable hypogastric mass. The scarcity of cases and the low number of studies carried out explains the lack of an evidence-based management strategy, but it seems that surgical treatment (open, laparoscopy or robot-assisted) represents the gold standard, while neoadjuvant and adjuvant chemotherapy or radiotherapy has a limited impact on overall survival. Since mucinous cystadenocarcinoma of urachal origin is a very uncommon pathological condition the differential diagnosis may be difficult and pathological investigations have to elucidate this disorder. It is worth mentioning the psychological impact on the patient in addition to the medical aspects. A rare condition is associated with heightened risk for mental health and psychosocial difficulties and this must be taken into account in the subsequent follow-up of the patient. In order to increase awareness of this rare entity we report a case of a 40-year-old male with a urachal adenocarcinoma who was treated surgically, with a favorable outcome. We also perform a brief literature review about this type of tumor.

## 1. Introduction

Urachus is an embryological remnant that connects the umbilicus and anterior wall of the bladder [1]. During the period of the fourth or fifth month of embryonic development, the urachus is gradually blocked as a fiber cord and enters the space between abdominal fascia and Retzius gap. The bladder descends into the pelvis at the same time. The urachus closes to become an umbilical median ligament in the embryonic evolution process [2]. Sometimes it undergoes an incomplete atresia and it can become the primary site of various lesions, such as cyst, fistula, neoplasia and diverticulum [3]. Up to 30% of the human urachus may not be completely occluded before birth and persist in adulthood. The urachal tube wall consists of three layers of structure, from inside to outside, represented by the transition epithelium, connective tissue and residual smooth muscle cells. Although urachal abnormalities are rare, the common complications are represented by tumors and infections. Umbilical discharge of mucus and urachal infection is more common in infants and children. Urachal cancer (UrC) is very rare, accounting for only 0.5–2% of bladder-associated malignancies and 0.01% of all cancers in adults. It has an insidious appearance, an aggressive behavior and a poor prognosis [4,5]. Ninety percent of urachal cancers are adenocarcinomas [6], believed to emerge from intestinal metaplasia of the epithelial component [7,8]. Non-glandular neoplasms can be urothelial, squamous cells, neuroendocrine, and mixed type [9]. Urachal carcinoma affects patients between 40 and 70 years of age, with a male predilection (male: female = 12:1) [10,11].

Frequently, patients present with microscopic or gross hematuria, occasionally with abdominal pain, dysuria or mucosuria. Rarely, some unusual urinary symptoms, such as pollakiuria, pyuria, recurrent urinary tract infection or umbilical discharge may occur. Systemic manifestations are often found in the elderly population, such us fever, weight loss, and nausea [12].

The diagnostic pattern may include imaging. Ultrasonography can expose an inhomogeneous mass located in the midline of the abdomen, sometimes associated with calcifications that are pathognomonic. Computed tomography (CT) scan or magnetic resonance imaging (MRI) are used in evaluating the tumor (staging, visceral involvement, lymphadenopathy or distant metastasis) [13,14]. A new investigation method, namely, Urinary UroVysion FISH, which has a sensitivity and specificity of 71.43% and 94.61%, respectively, is used in diagnosis of urachal carcinomas and seems to differentiate UrC from a benign urachal cyst [15].

The stage of the disease decides the management of the treatment and oncological outcomes. Surgery is the gold standard for localized tumoral mass. Excision of the urachus and the umbilicus, and partial/radical cystectomy, with extended bilateral pelvic lymphadenectomy (PLND), are routinely performed. The target in surgical management of UrC is to obtain negative surgical margins, particularly after partial cystectomy [12,13]. Regularly, the preferred surgical approach is the open one, but laparoscopy is a safe and available option [16]. Nevertheless, the role of bilateral pelvic lymphadenectomy is still controversial in not improving overall survival (OS) and is associated with a high complication rate and only 17% of lymph node positivity [12,17].

Perioperative chemotherapy has a limited role, especially in advanced stages when the patient has a poor prognosis. Neoadjuvant chemotherapy could be recommended in unresectable tumors, but only if the response may lead to surgical treatment with negative margin resection [18]. Radiotherapy has been used in only 10% of UrC cases (Surveillance, Epidemiology, and End Results (SEER) statistics) [19]. The available evidence on it indicates that it is inefficient as a local treatment [8].

Regarding urachal tumors, they are associated with a poor prognosis and with a five-year survival rate of between 9 and 50 % [10,20,21]. The scarcity of cases and the low number of studies carried out explains the lack of an evidence-based management strategy.

### Justification of the Case Report

Mucinous cystadenocarcinoma of urachal origin is a very uncommon pathological condition, so the differential diagnosis may be difficult and pathological investigations have to elucidate this disorder. Due to its rarity, there is some disagreement in the literature about the diagnostic criteria, the staging system and the best treatment options. Previous studies about UrC are insufficient, and the optimal therapeutic management is still a matter of debate. In order to increase awareness of this rare entity we report a case of a 40-year-old male with only gross hematuria as a symptom and with a large nonspecific intra-abdominal tumoral cystic mass: UrC (23 × 9 × 8 cm), adherent to the anterior abdominal wall and to the urinary bladder. We also perform a brief literature review about this type of tumor.

## 2. Case Report

A 40-year-old male patient presented to our hospital with a one-week history of lower urinary symptoms (gross hematuria) and lower abdominal pain. He denied nausea, vomiting, constipation, diarrhea, or any other associated symptoms. The patient was otherwise healthy with no prior surgical or medical interventions and no history of tobacco, alcohol or drugs use. Suprapubic ultrasonography performed in another hospital showed an enlarged cystic intra-abdominal mass in his lower abdomen. The mass, which was measured as 80/90/120 mm, possessed a well-defined boundary, regular shape and several separations nearing the bladder. The patient was admitted to the General Surgery department of the Military Emergency Hospital, Iași, for diagnosis and treatment.

### 2.1. Clinical Findings

Physical examination revealed a large, mobile mass in the hypogastric region. Laboratory studies (blood analysis, including tumor markers of free and total prostate specific antigen (PSA)), tumor markers (carcinoembryonic antigen (CEA) and cancer antigen (CA 19-9)) were unremarkable.

### 2.2. Imaging Findings

A contrast computerized tomography (CT) scan of the chest, abdomen and pelvis revealed parietal thickening of the antero-superior bladder wall (40 mm), with enhanced contrast. A heterogeneous, hypodense mass was located on the topography of the communication with a voluminous anterior bladder diverticulum (9 cm). The tumor had developed partly in the bladder, partly in the diverticulum and infiltration of surrounding tissue of the bladder near to the tumoral mass was identified.

No abnormal enhancement was identified. Small calcifications were seen in the lower part of the cystic mass. There was no retroperitoneal lymphadenopathy, ascites, or intestinal mechanical obstruction (Figure 1A–C).

Cystoscopy detected mucosal erosion and excluded a potential invasion of the bladder. Furthermore, it revealed no bladder diverticulum. Pelvic Magnetic Resonance Imaging (MRI) showed a large cystic mass confined to the pelvis measuring 75/91/127 mm (AP/TR/CC). The complex cystic lesion extended from the umbilicus to the antero-superior dome of the bladder, on the trajectory of the medial umbilical ligament. The tumoral mass extended anteriorly to the plane of the rectus abdominal muscles, and the postero-lateral came into contact with some intestinal loops and a portion of the sigmoid colon. In the caudal portion the antero-superior wall of the bladder was imprinted by a parietal infiltrative nodular lesion (not exceeding the mucous layer), in hyposignal T1 and T2, with several areas in hypersignal T2 and STIR, without diffusion restriction and intense contrast and size of 27/31/29 mm (AP/TR/CC).The aspects might suggest an urachal cyst with milky transformation in the portion of the bladder dome. There were several nodes with a short infracentrimetric axis located external iliac and bilateral inguinal (Figure 2A–C).

Radiologically, a diagnosis of urachal tumor with unknown malignant potential was put forth.

### 2.3. Operative Findings

The patient underwent a surgical radical resection through a suprapubic midline incision. Laparotomy revealed a large intra-abdominal cystic mass adherent to the anterior abdominal wall and to the urinary bladder (Figure 3A,B). We performed radical resection of the tumor en bloc with the umbilicus and partial cystectomy (Figure 4A,B). The cystic mass measured approximately 200/90/80 mm and contained mucinous material.

### 2.4. Pathological Findings

Macroscopically, the mass weighed 583 g and measured 23 × 9 × 8 cm. It was covered by a thin membrane. The cut surface showed a cystic mass filled with yellow-greenish semisolid material.

Microscopically, the fragment of the bladder wall was lined with urothelium without dysplastic changes. The wall identified cavity space lined with urothelium without significant changes, located in an area of fibrosis with chronic inflammatory infiltrate. Morphological aspects related to clinical data may correspond to the urachal orifice.

The cystic space was characterized by fibrosis and a sclero-hyalinosis wall, chronic inflammatory infiltrate and calcifications, and lined with intestinal mucinous epithelium with low-grade dysplasia, but also with high-grade areas, frequently associated with mitosis. There were also places of intraparietal extracellular mucus, lined with mucinous epithelium with high-grade dysplasia. The cystic wall had adipose tissue or smooth muscle bundles in the periphery.

The described intra-cystic area was represented by sclero-hyalinosis with important calcifications and bone metaplasia. It was noticed on a slope, in the areas of sclero-hyalionosis, that there was extracellular mucus in which nests of cells with important cytonuclear atypia and mitosis were identified, an aspect that could correspond to a minor component of mucinous adenocarcinoma.

Morphological and immunohistochemistry (IHC) aspects correlated with imaging data argued in favour of a mucinous cystadenocarcinoma of urachal origin, possibly developed on a potentially malignant mucinous cystic tumor. Bladder intraparietal arrangement and excision margins on the examined sections located in normal adipose connective tissue corresponded to local evolution (Sheldon Stage IIIA). No tumoral involvement of the bladder was found on the examined tissue section.

Immunohistochemical stains expressed CK20, was positive for CK7, SATB2 and CK34𝛽E12 and presented diffuse membrane staining for *β*-catenin. These IHC results supported the diagnosis of urachal origin.

### 2.5. Follow-Up

The patient’s urinary symptoms were relieved, abdominal drainage was removed on the sixth day and he was discharged on the 7th postoperative day after an uneventful stay. The urethral catheter was removed after 21 days and no complications were recorded after the patient’s release. The patient was subsequently referred to a medical oncologist who decided to perform periodic imaging evaluation without further adjuvant treatment. The patient was evaluated by thoraco-abdomino-pelvic computer tomography after 3 months from surgery. The scan did not identify any signs of local recurrence or pathological lymphadenopathies (Figure 5).

## 3. Sources of Information

Using the Medical Subject Headings MeSH term “urachal” and “urachal carcinoma”, we performed a PubMed literature search for randomized controlled trials (RCTs), systematic reviews, observational studies, series of cases studies and case reports from the earliest possible date to May 2022. Articles and their reference lists, published in English, were analyzed for other relevant articles. Searching PubMed for the term “urachal”, 1663 articles were found, but when the term “carcinoma” was added, only 365 articles returned. We reviewed the title and the abstract of the articles and the full text of 98 articles, to extract data on incidence, sex ratio, diagnosis, etiology, associated abnormalities or treatment. Ninety-two of all the articles found were quoted.

## 4. Discussion

Primary urachal adenocarcinoma is a rare entity, first described by Hue and Jacquin in 1863 and responsible for only 10% of primary bladder adenocarcinoma [22,23]. Less than 400 cases have been reported in the literature since it was first described [8]. Albeit, there is still disagreement regarding the origin of UrC, the glandular metaplasia basis and the origin of glandular intestinal epithelial cells are the most commonly accepted premises to elucidate the dominance of adenocarcinoma of the UrC [4,24]. UrC is more frequent in the male population and commonly occurs between the ages of 50 and 60 years old [24,25]. In the current study, the most common presenting symptom was gross hematuria, which agreed with the previous reports [9].

UrC is often perceived as a poor prognosis malignancy, but, corresponding to SEER and RKI records, the relative five-year CSS rate was 54.8% in Germany and 64.4% in the USA [26]. While early diagnosis is mandatory for patients with UrC, no clear risk factors were identified. Multiple parameters were evaluated, such as age, sex, tumor grade, tumor size, LN status, Sheldon stage, Mayo stage, the presence of distant metastasis, surgical intervention, positive surgical margin and anatomopathological type of the tumor [16,27,28]. The meta-analysis conducted by Szarvas et al. uncovered that criteria, such as Sheldon stage 4IIIB, Mayo stage 4II, presence of LN or distant metastases, positive surgical margin, and ECOG performance status, were independent risk predictors [12]. Prior articles revealed that the histologic tumor stage and negative surgical margins were the most eloquent predictors of survival for UrC [8,9,24]. Clinical manifestation, biological tests and medical imaging may help set the diagnosis, but pathological evaluation after surgery provides the important details to establish the diagnosis of certainty and the degree of malignancy. The unusual case of a 40 year old male with a tumoral mass and hematuria, without other associated conditions, who required surgical excision has been reported above. His postoperative evolution was positive at the 7th day and at three-month follow-up, although the median time of recurrence after the resection of the primary tumor is reported to be 29 months [29].

Almost 70% of urachal adenocarcinomas are mucin-producing tumors and, as happened in our case, display calcifications [30]. Abdominal suprapubic pain and dysuria may appear, but hematuria is the most common symptom and the disease is usually advanced when this symptom appears. Frequent metastatic sites include the pelvic lymph nodes, peritoneum, bones and lungs, but survival is not correlated with the site of the metastatic disease [31]. Although the only discomfort and reason for presenting our patient to the doctor was macroscopic hematuria, evaluation of CT, MRI and surgery did not reveal an advanced stage of the disease.

A computer tomography detail of urachal carcinoma is a midline mass antero-superior to the dome of the bladder with low-attenuation components represented by mucin [32]. In 50% to 70% of cases, peripheral calcifications (punctate, stippled, or curvilinear) in the soft-tissue attenuation mass occur and are pathognomonic for urachal adenocarcinoma [32,33,34]. MRI is an excellent staging tool. Due to the presence of mucin within the tumor, increased signal intensity is seen on T2-weighted spin-echo [35]. Both CT and MR imaging are useful for demonstrating intra- and extra-vesical extension of the tumor, as in our case. Cystoscopy allows visualization of the location, size and degree of invasion, offers the possibility of biopsies and, in our case, excluded the presence of a diverticulum. Infection of the urachal remnants may mimic urachal carcinoma, resulting in challenges for imaging diagnosis; therefore, in unclear cases, biopsies are recommended. In general, hematuria and calcification are more likely to be urachal carcinoma, while in the female gender, abdominal pain and thickening of adjacent bladder wall are more likely to be infections [11]. Additionally the typical extension of urachal carcinomas along the Retzius space helps differentiate them from vesical carcinomas [29].

Tumor markers have become the key support for the diagnosis of various neoplastic lesions, but not all have sensitive markers. Considering the low incidence of urachal tumors, the literature has not identified a sensitive tumor marker.

The largest cohort registered was by Siefker-Radtke; they found elevated (>3 ng/mL) CEA serum levels in 59% of patients with urachal adenocarcinomas (median: 36 ng/mL), which also decreased in response to chemotherapy, suggesting the potential utility of CEA testing in follow-up surveillance of the patients [29,36]. Elevated serum levels of CA19-9 and CA125 were reported in many studies (in 50.8% (31/61) and 51.4% (19/37) of patients, for example), but the markers are not specific [35,36,37,38,39,40,41,42,43,44,45,46,47,48,49,50,51,52,53,54,55,56,57,58,59]. In the presented case these tumor markers had normal values.

Other serum biomarkers described in low case numbers of urachal adenocarcinomas include lactate dehydrogenase (LDH) [41], cancer antigen 15-3 (CA15-3) [29,49,60], AFP [4,24,32,47,48,49] and neuron-specific enolase (NSE) [60,61,62,63,64]. Measurement of serum biomarkers might be effective in the follow-up and disease monitoring of urachal carcinoma.

Amin et al. and Paner et al. have created a classification system to help maintain consistency in naming the epithelial neoplasms of the urachus and more specifically the mucinous cystic tumors (Table 1) [65,66].

A set of criteria was published in the 2016 World Health Organization (WHO) for the diagnosis of urachal adenocarcinoma and they are [67]:Location of the tumor in the bladder dome and/or anterior wallEpicenter of carcinoma in the bladder wallAbsence of widespread cystitis cystica and/or cystitis glandularis beyond the dome and anterior wallAbsence of a known primary tumor elsewhere.

There are a few staging systems for urachal carcinomas (Sheldon, Mayo, TNM staging). The universally accepted one, and the most accurate, is the staging proposed by Sheldon (Table 2) [10,32], then the Mayo staging system (Table 3) [33]. According to this, the presented case was classified as Sheldon Stage IIIA.

Biomarkers for differential diagnostic purposes are required given the overlapping histopathological features of adenocarcinomas of urachal and primary bladder origin and from different sites (colorectal, ovarian, appendix) [46,54,68,69,70]. The differential diagnostic problems with major impact on therapeutic decisions may be categorized as follows:5.Differentiating between invasion/metastasis of colorectal adenocarcinomas and urachal adenocarcinomas6.Distinguishing urachal adenocarcinomas from those of primary bladder origin7.Identification of the origin of a mucinous adenocarcinoma of unknown primary sites is also important because urachal adenocarcinomas regularly metastasize to various organs, such as the bone, lung, and liver [71].

The immunohistochemical markers most often engaged in the work up of adenocarcinomas of different sites usually include Cytokeratin 20 (CK20) and CK7. The urachal adenocarcinoma is generally positive for CK20 and CK7 in 60% of the cases, and for CK34𝛽E12 in 66% of the cases, but only very focally [66,72]; nuclear staining with 𝛽-catenin occurs in 6% [66], normally showing only cytomembranous staining. In the case reported on herein, IHC stains expressed CK20, were positive for CK7, SATB2 and CK34𝛽E12, and presented diffuse membrane staining for *β*-catenin.

In the study conducted by Henning Reis et al., they concluded that CK20, havingan overall positive rate of 97%, and CK7 displayed a pooled reactivity rate of 51%, compared to considerably lower rates in colorectal cancer (0–38%) [71].

β-Catenin is a protein involved in cell-cell adhesion and gene transcription regulation and, as a biomarker, can be found in the majority of cases [73]. Nuclear β-Catenin reactivity was detected in a small rate of primary bladder and urachal adenocarcinoma [74,75]. Unfortunately, β-Catenin is no use in the differentiation of primary bladder from urachal adenocarcinomas because both entities exhibit β-Catenin staining characteristics [75].

Useful biomarkers of urachal adenocarcinomas, like alpha-methylacyl-CoA racemase (AMACR, p504s), CD15 (Leu-M1), carcinoembryonic antigen (CEA), CK34βE12 (high-molecular weight cytokeratin), GATA binding protein 3 (GATA3), mucin 2 (MUC2), and mucin 5AC (MUC5AC), should be taken into consideration [71]. Furthermore, CK34βE12 is more frequently positive (67%) in urachal adenocarcinomas, while being irregularly expressed in primary bladder or colorectal adenocarcinomas [36,76,77,78,79]. A comparable distribution was detected for MUC2 and MUC5AC with high positivity rates in urachal adenocarcinomas (100% and 92%) and lower rates in colorectal and primary bladder adenocarcinomas [43,46,50,59,68,69,80,81,82].

The differential diagnostic between the various tumors entities is enlightened by immunohistochemical staining of a panel of antibodies. This process also includes the use of further antibodies in addition to the core panel.

The recommended treatment for nonmetastatic cases is surgery, but no standard surgical treatment has been recommended until now. In localized disease, primary bladder adenocarcinomas are usually treated with complete cystectomy, while urachal adenocarcinomas mostly require partial cystectomy with en bloc removal of the umbilical ligament and umbilicus (radical versus partial cystectomy) and have a significantly different impact on quality of life [8,9]. Partial or radical cystectomy has similar oncologic results, but partial cystectomy should be preferred because it ensures a better quality of life and has less complications. Complete tumor resection (negative surgical margins) is of great importance in partial cystectomy; postoperative care, bladder drainage, carefully chosen antibiotic therapy and psychological support of the patient must be carried out in a multidisciplinary team [10,34,83]. Thus, en bloc resection with complete removal of urachal remnant and the umbilicus should be the surgery performed for prolonged survival [34]. Lymph node resection was not a predictor of overall survival [10]. Laparoscopic or robotic surgery may be considered for selected cases [21,37,44,70].

The role of neoadjuvant or adjuvant chemotherapy or radiotherapy (R1 excision) is still to be proven [45,53]. Various protocols have been studied, some based on cisplatin, and others based on fluoro-uracil [84]. Due to the rarity of urachal cancer, there is no universally accepted targeted therapy, and, therefore, it shares the molecular profile and treatment of colorectal carcinoma [10,85]. Further research and long-term observational studies are needed.

Psychological challenges are related to medical aspects of the disease. More precisely, the lack of knowledge of patients about the medical condition, uncertainty about the future, evolution, disease progression and treatment. Attitude of the medical multidisciplinary team involved should guide the patient in managing his or her medical and social situation [86].

## 5. Conclusions

Urachal adenocarcinoma is a rare condition with a poor prognosis due to locally advanced or metastatic disease at the time of diagnosis, especially because most patients are asymptomatic. The most common symptoms are hematuria and the presence of a palpable hypogastric mass, such as in the case presented. Surgical treatment represents the gold standard, while adjuvant chemotherapy or radiotherapy has a limited impact on overall survival. Besides the rarity of the condition in the general population, another particularity of the case is that rare diseases lead to a negative psychological impact on the patient. People with rare diseases may face challenges different from those with frequent medical conditions.

## Figures and Tables

**Figure 1 diagnostics-12-01892-f001:**

(**A**–**C**) Heterogeneous, hypodense mass develop partly in the bladder (**A**,**B**); infiltration of the surrounding tissue (**C**).

**Figure 2 diagnostics-12-01892-f002:**
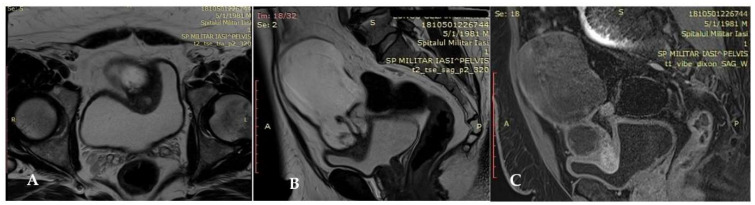
(**A**–**C**) Cystic lesion extended from umbilicus to antero-superior dome of the bladder, on the trajectory of the medial umbilical ligament (**A**,**B**) and with involvement of the antero superior bladder wall (**C**).

**Figure 3 diagnostics-12-01892-f003:**
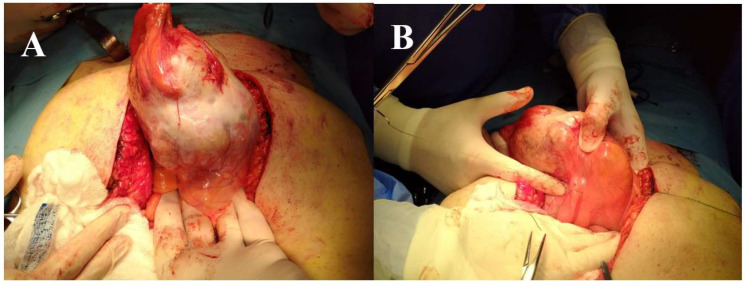
(**A**). Macroscopic aspect of tumor: the suprapubic mass consisted of solid and cystic lesions. (**B**). Tumoral cystic mass adherent to antero-superior wall of the bladder.

**Figure 4 diagnostics-12-01892-f004:**
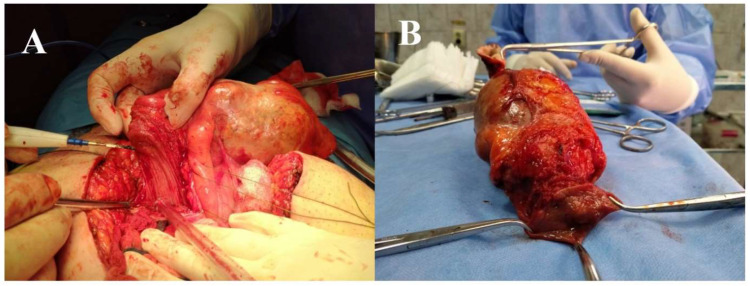
(**A**). Intraoperative view: The opening of the bladder in order to perform partial cystectomy (tumor invasion). (**B**). Radical resection of the tumor en bloc with the umbilicus and partial cystectomy.

**Figure 5 diagnostics-12-01892-f005:**
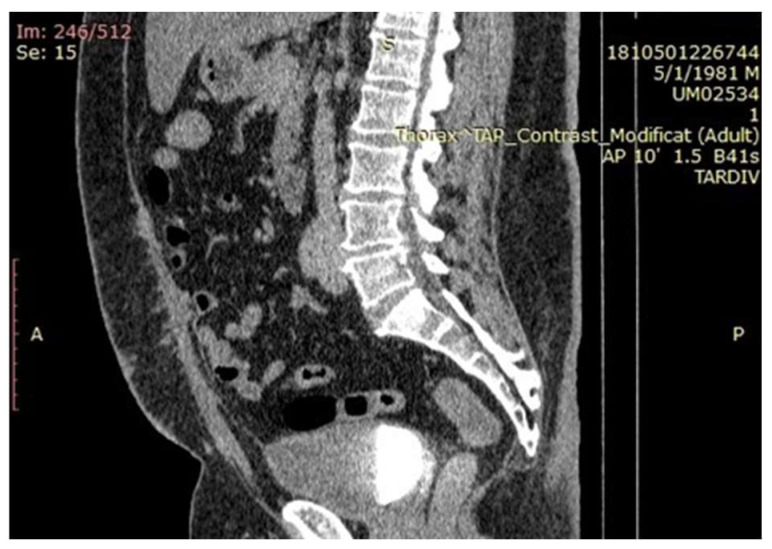
CT examination revealed no local tumoral recurrence or adenopathies.

**Table 1 diagnostics-12-01892-t001:** Classification of epithelial neoplasms of urachal origin with emphasis on the cystic mucinous neoplasms, modified from Paner et al., 2016, & Amin et al., 2014 [65,66].

Glandular Neoplasms
(I) Adenoma
(II) Cystic mucinous neoplasms:
(a) Mucinous cystadenoma (cystic tumor with a single layer of mucinous columnar epithelium, with no atypia)
(b) Mucinous cystic tumor of low malignant potential (cystic tumor with areas of epithelial proliferation, including papillary formation and low-grade atypia/dysplasia)
(c) Mucinous cystic tumor of low malignant potential with intraepithelial carcinoma (cystic tumor with significant epithelial stratification and unequivocal malignant cytological features and often with stroma-poor papillae and cribriform pattern)
(d) Mucinous cystadenocarcinoma with microinvasion (stromal invasion <2mm and comprising <5% of the tumor)
(e) Frankly invasive mucinous cystadenocarcinoma (stromal invasion that is more extensive than 2 mm and 5%)
(III) Non-cystic adenocarcinoma
Non-glandular neoplasms
(I) Urothelial neoplasm
(II) Squamous cell neoplasm
(III) Neuroendocrine neoplasm
(IV) Mixed-type neoplasm
NOS: not otherwise specified.

**Table 2 diagnostics-12-01892-t002:** Urachal cancer staging system as defined by Sheldon et al. [32].

Stage Definition
I Urachal cancer confined to urachal mucosa
II Urachal cancer with invasion confined to urachus itself
IIIA Local urachal cancer extension to the bladder
IIIB Local urachal cancer extension to the abdominal wall
IIIC Local urachal cancer extension to the peritoneum
IIID Local urachal cancer extension to viscera other than the bladder
IVA Metastatic urachal cancer to the lymph nodes
IVB Metastatic urachal cancer to distant sites

**Table 3 diagnostics-12-01892-t003:** Urachal cancer staging system as defined by the Mayo Clinic [33].

Stage Definition
I Tumors confined to the urachus and or bladder
II Tumors extending beyond the muscular layer of the urachus and/or the bladder
III Tumors infiltrating the regional lymph nodes
IV Tumors infiltrating non regional lymph nodes or other distant sites

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
