# Peer review of "Urachal Carcinoma, An Unusual Possibility of Hematuria; Case Report and Literature Review"

_diagnostics, 2022, doi:10.3390/diagnostics12081892_

Round 1

Reviewer 1 Report

The case report by author Calin et al and group entitle “Urachal carcinoma, an unusual possibility of hematuria Urachal cancer” is a very unusual and rare type of cancer, the author did perform a computerized tomography (CT) scan of the chest, abdomen, and pelvis the image showed that the tumor develops partly in the bladder, partly in the diverticulum; infiltration of surrounding tissue of the bladder near to the tumoral mass is identified parietal thickening of the anterosuperior bladder wall. The present case was also supported by IHC of CK20, was positive, for CK7, SATB2, and CK34?E12, and presented diffuse membrane staining for ?-catenin. Moreover, the author supported the data with an MRI scan that showed a large cystic mass confined to the pelvis measuring 75/91/127 mm. The Cystic lesion extended from the umbilicus to the antero-superior dome of the bladder, on the trajectory of the medial umbilical ligament. The follow-up study after 3 months of surgery did not show any signs of local recurrence or pathological lymphadenopathies. Overall the report is very well presented with all supporting data such as IHC, MRI and follow-up study.

The line 153 change …..ant to and (typo mistake) 

Author Response

Dear reviewer,

Thank you very much for evaluating our manuscript and for your kind remarks. 

Reviewer 2 Report

General comment

The manuscript entitled “Urachal carcinoma, an unusual possibility of hematuria” report the case of a 40 years old patients with a large mass which was an urachal adenocarcinoma, treated surgically. The illustration of the case is interesting and could enrich the related literature. However, different corrections have to be made before considering the manuscript suitable for publication.

I suggest to revise English grammar and language, improve the part related to the review of literature and clarify some aspect of the management of the case.

Detailed suggestions are reported below:

-          Major issues

TITLE

I would modify the title report also the review of literature.

INTRODUCTION

51-56: Albeit the characteristics of urachal cancer, as well its epidemiological data, are reported, you could improve this section briefly adding the main symptoms, the diagnostic pattern and the therapeutic options.

60: The justification of the case report seems superfluous. You could simply state why you report this case and what is your objective. Considering the rarity of the condition, the increased mass of the tumor of your case and adding a review of the literature on the topic, as justification is more than enough.

CASE REPORT

94: If available, report the images of the cystoscopy.

144-148: It is not clear if the bladder was involved.

152: Please clarify if drainages and urinary catheter was placed. If so, report also the time of removal and discharge, as well as other potential interesting data on the post-operatory (complications, time to canalization ecc)

DISCUSSION

The discussion need to be improved. Firstly, you have to state how many other reported cases are described in the literature, in order to provide a background for your case report. Secondly, the pathogenesis, as well as risk factors, genetics and all other possible factors involved have to be reported. Thirdly, I don’t see a proper review of the literature, as cited in the abstract. Report other similar cases, focusing on differences and similarities with your case. Lastly, albeit reporting the classification of urachal cancer is important and on point, you could cull the excessive data and focus on the previous reported issue.

-          Minor issues

ABSTRACT

Revise the abstract as it seems redundant in the second part.

INTRODUCTION

48: If you say that urachal cancer is rare, how could you write that tumors and infections are common? Please clarify.

CASE REPORT

69: Any data on tobacco and alcohol? Other drugs taken by the patient?

SOURCES OF INFORMATION

This section could be moved to the discussion.

Author Response

Dear reviewer,

Thank you very much for evaluating our manuscript. We tried to take advantage of your suggestions to improve the quality of the manuscript. The changes we made are highlighted in red.

Major

  1. Q: I would modify the title report also the review of literature.
    R: We modified the title as you suggested: Urachal carcinoma, an unusual possibility of hematuria; case report and literature review
  2. Q: 51-56: Albeit the characteristics of urachal cancer, as well its epidemiological data, are reported, you could improve this section briefly adding the main symptoms, the diagnostic pattern and the therapeutic options. R: We added a paragraph with the main symptoms, the diagnostic pattern and the therapeutic options (line 58-84).
  3. Q: 60: The justification of the case report seems superfluous. You could simply state why you report this case and what is your objective. Considering the rarity of the condition, the increased mass of the tumor of your case and adding a review of the literature on the topic, as justification is more than enough. R: We modified this section (line 89-98).
  4. Q: 94: If available, report the images of the cystoscopy. R: Cystoscopy detected mucosal erosion and excluded a potential invasion of bladder, but unfortunately the images could not be stored and were not saved. 
  5. Q: 144-148: It is not clear if the bladder was involved. R: Morphological and immunohistochemistry (IHC) aspects correlated with imaging data argue in favour of a mucinous cystadenocarcinoma of urachal origin possibly developed on a potentially malignant mucinous cystic tumor. Bladder intraparietal arrangement and excision margins on the examined sections located in normal adipose connective tissue correspond to a local evolution (Sheldon Stage IIIA). No tumoral involvement of the bladder was found on the examinated tissue section. (line 177-182)
  6. Q: 152: Please clarify if drainages and urinary catheter was placed. If so, report also the time of removal and discharge, as well as other potential interesting data on the post-operatory (complications, time to canalization ecc). R: Patient’s urinary symptoms were relieved, abdominal drainage was removed on the sixth day and he was discharged on the 7th postoperative day after an uneventful stay. The urethral catheter was removed after 21 days and no complications were recorded after the patient's release. The patient was subsequently referred to a medical oncologist who decided to perform periodic imaging evaluation without further adjuvant treatment. The patient was evaluated by thoraco-abdomino-pelvic computer tomography after 3 months from surgery. The scan did not identify any signs of local recurrence or pathological lymphadenopathies. (line 187-194). 
  7. Q: The discussion need to be improved. Firstly, you have to state how many other reported cases are described in the literature, in order to provide a background for your case report. Secondly, the pathogenesis, as well as risk factors, genetics and all other possible factors involved have to be reported. Thirdly, I don’t see a proper review of the literature, as cited in the abstract. Report other similar cases, focusing on differences and similarities with your case. Lastly, albeit reporting the classification of urachal cancer is important and on point, you could cull the excessive data and focus on the previous reported issue. R: We have improved the Discussion section taking into account your recommendations and adding 15 new references. (line 208-228).

Minor

  1. Q: Revise the abstract as it seems redundant in the second part. R: We revised the abstract.
  2. Q: 48: If you say that urachal cancer is rare, how could you write that tumors and infections are common? Please clarify. R: Althought the urachal  abnormalities are rare, the common complication are represented by tumors and infections. We specified this in text.
  3. Q: 69: Any data on tobacco and alcohol? Other drugs taken by the patient? R: The patient was otherwise healthy with no prior surgical or medical interventions and no history of tobacco, alcohol or drugs use (see text).
  4. Q: SOURCES OF INFORMATION - This section could be moved to the discussion. R: We decided to leave it like that, asking for the editors' recommendation whether to modify or not.

Thank you again for your time and recommendations.

Kind regards.

Round 2

Reviewer 2 Report

The manuscript was improved according to the suggestions. No further corrections are required.